# Finding Fingerprints of Out-Of-Distribution Failures from In-Distribution Geometry

Chi-Ning Chou[1,2]     Artem Kirsanov[3]     Yao-Yuan Yang[4]     SueYeon Chung[1,2,5]

## Abstract

Generalization, the ability to perform well outside the training context, is a hallmark of both biological and artificial intelligence. A key challenge is to anticipate potential failure modes at evaluation time using only the information available at training time. In this work, we study image classification tasks where the image classes differ between training and evaluation, and ask whether failure in such out-of-distribution (OOD) generalization can be predicted by analyzing the representations (i.e., feature vectors) of in-distribution (ID) training data. Across architectures, network sizes, training parameters, optimization algorithms, and datasets, we find that conventional metrics fail to robustly predict OOD generalization, while task-relevant geometric signatures of ID representations strongly correlate with it. Specifically, networks tend to generalize poorly to new image classes when the dimensionality of ID object manifolds are more compressed in the feature space. Our results highlight representational geometry as a promising lens for mechanistic interpretability and robustness, with potential implications for comparing biological and artificial neural systems.

## 1  Introduction

In biology and medicine, *biomarkers*—such as high cholesterol, altered hormone levels, or atypical brain activity—can indicate future health risks, like heart disease or neurological disorders, even when outward symptoms are absent [1, 2, 3, 4]. As deep neural networks (DNNs) become increasingly integrated into real-world applications, it is equally important to anticipate their failure modes—particularly in out-of-distribution (OOD) settings where training and deployment environments may differ significantly [5]. While a model may perform well on in-distribution (ID) training and test sets, such performance does not guarantee robustness under a shift of the underlying data distribution [6, 7, 8]. Motivated by this analogy, we ask: can we foresee a model's failure mode by inspecting its internal representations – even before the failure occurs? Concretely, can we design "biomarkers" from the neural activity patterns evoked by ID training data that signal potential failures in OOD generalization?

### 1.1  Our contributions

In this work, we systematically investigate a broad class of deep networks trained or fine-tuned on ID data (e.g., CIFAR-10 [9]) to test whether geometric and statistical properties of their internal representations can predict OOD generalization performance on a disjoint dataset (e.g., CIFAR-100 [9])(Figure 1a,b). We evaluated a range of model architectures (ResNet [10], VGG [11], Efficient-

---

[1]Center for Computational Neuroscience, Flatiron Institute. Contact: cchou@flatironinstitute.org
[2]Department of Physics, Harvard University.
[3]Program in Neuroscience, Harvard University.
[4]Google DeepMind. Worked in an advisory capacity.
[5]Applied Mathematics, Harvard University.

Preprint.

Net [12], MobileNet [13], DenseNet [14]), depths, optimization algorithms (SGD, AdamW [15]), and training hyperparameters (initial learning rate, weight decay). Our key findings are:

- Different training hyperparameters can lead to markedly different OOD performance, despite similar ID train and test accuracy.

- Conventional performance metrics (e.g., train/test accuracy) and statistical measures (e.g., sparsity, covariance) of ID representations are weakly predictive of OOD performance (Figure 1c, top).

- In contrast, several geometric properties of ID (both train and test data) feature representations — especially task-relevant ones — show strong correlation with OOD performance. In particular, the OOD performance drops when the ID object manifolds (i.e., the point clouds of feature vectors from the same image class) are more compressed (Figure 1c, bottom).

These results suggest that ID representational geometric measures can serve as early indicators of potential OOD failure modes in neural networks, offering a path toward better interpretability and robustness. Beyond engineering implications, these measures may also act as intermediate descriptors to uncover the mechanisms of generalization, or as "biomarkers" for comparing biological and artificial neural representations under domain shift.

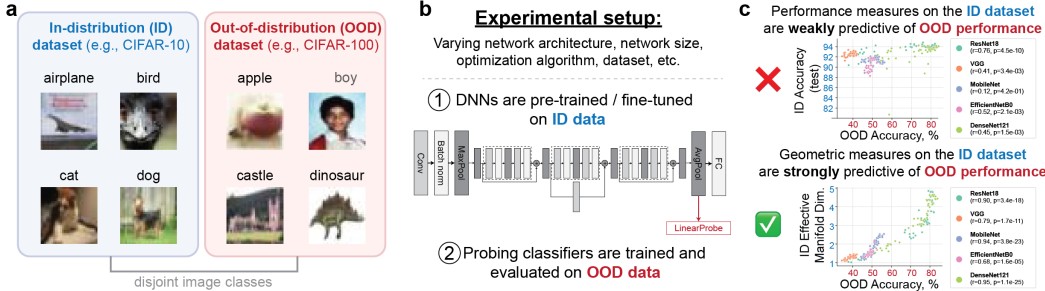

Figure 1: Schematic illustration. **a**: We consider image classification problem with an in-distribution (ID) dataset and an out-of-distribution (OOD) dataset with disjoint image classes. **b**: We pre-trained / fine-tuned DNNs on the ID dataset and evaluated the OOD performance as the validation accuracy of a linear probe trained on the OOD dataset. **c**: Conventional performance (e.g., train/test accuracy) and statistical (e.g., sparsity, covariance) measures on the ID dataset are weakly predictive of OOD performance, while some geometric measures (e.g., dimensionality of object manifolds) can robustly predict failures in OOD generalization.

## 1.2 Related work

A growing body of work suggests that properties of internal representations in deep networks can serve as indicators of generalization performance. In particular, both statistical properties of activations — such as sparsity, covariance, and inter-feature correlations [16, 17] — and geometric measures of object manifolds [18, 19, 20] have been shown to be predictive in standard ID settings. For example, networks that generalize well often exhibit low intrinsic dimensionality in their final-layer representations, and such compactness has been shown to correlate with test accuracy in image classification tasks [18]. However, in the context of OOD generalization, the picture is less clear: the relationship between representation structure and generalization often depends on the nature of the distribution shift. For instance, in video and image classification tasks, diverse and decorrelated representations have been associated with better OOD robustness [17], whereas collapsed or overly aligned representations may signal brittle generalization. These observations echo findings in neuroscience, where high-dimensional yet smooth population codes in mouse visual cortex have been linked to generalization across stimulus conditions [4]. These results motivate further inquiry into how representational structures connect to generalization performance in OOD settings.

## 2   Methods

We adopt an experimental design in [20] where DNNs are trained on an ID image dataset and OOD performance is evaluated on a different dataset with a disjoint set of classes. We analyze two primary experimental settings (training from scratch and fine-tuning).

**Training regimes.**   First, in a broad hyperparameter sweep, we trained several DNN architectures (e.g., ResNet, VGG) from scratch on CIFAR-10. For each architecture, we performed a grid search over initial learning rates and weight decay values using both SGD and AdamW optimizers. Second, to study the training dynamics, we fine-tuned an ImageNet-pretrained ResNet50 on a subset of the Stanford Cars dataset of variable size, saving 25 intermediate checkpoints during training across 10 different learning rates. Full implementation details, including hyperparameter ranges and dataset splits, are provided in Appendix A.

**OOD evaluation via linear probing.**   To assess the OOD generalization of learned representations, we adopt a linear probing framework [21, 20]. After ID training, the network's feature extractor was frozen. A new linear classifier was then trained on top of these features using the OOD dataset. The test accuracy of this linear probe served as our measure of OOD performance (Figure 1b).

**ID measures for detecting failures in OOD generalization.**   We evaluated three categories of "biomarkers" computed from the feature vectors from the penultimate layer of ID data.

1. Performance Measures: Standard training and test accuracy on ID data
2. Statistical Metrics: Activation sparsity, feature covariance and pairwise feature distances
3. Geometric Measures: Participation ratio and task-relevant geometric factors from the GLUE framework, such as effective dimension and radius of class-specific object manifolds [22, 23]. Detailed definitions are provided in Appendix B.

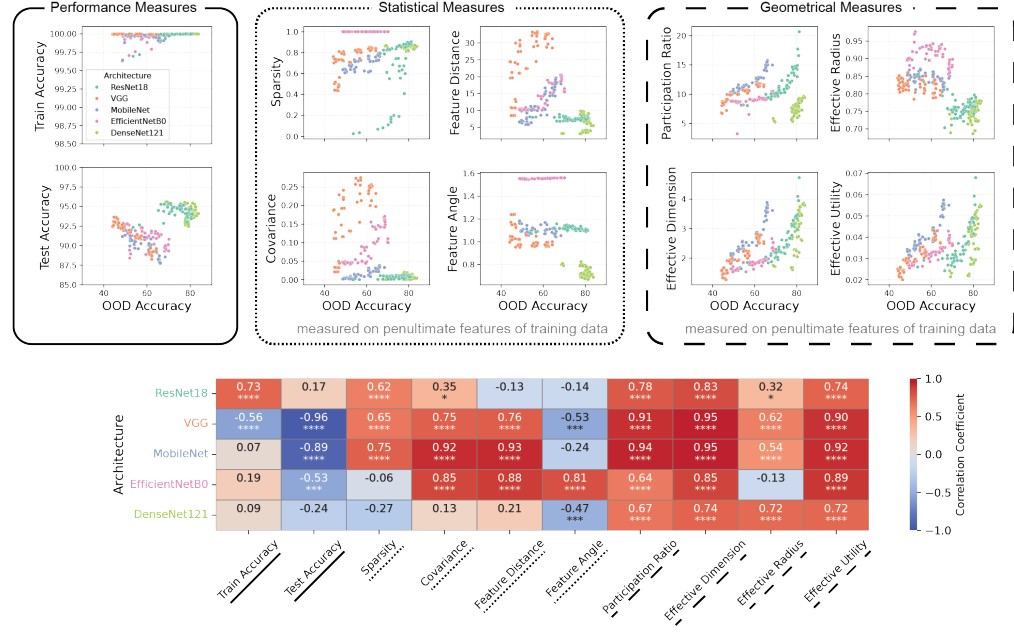

Figure 2: Results on four feedforward DNNs trained on CIFAR-10 using SGD with varying initial learning rate, weight decay, and seed. The OOD dataset is CIFAR-100. Numbers indicate Pearson $r$; asterisks denote significance (* : $p \leq 0.05$; **** : $p \leq 0.01$; *** : $p \leq 0.001$; **** : $p \leq 0.0001$)

## 3   Results

We find that models with similar ID accuracy can exhibit vastly different OOD performance. This variation, however, is not random; it is consistently predicted by the geometric properties of the ID representations.

### 3.1 Geometric "biomarkers" are predictive across architectures

First, we trained different vision architectures (ResNet, VGG, etc) on CIFAR-10 and evaluated OOD performance on CIFAR-100. Each point in our analysis corresponds to a model trained with a unique set of hyperparameters. As summarized in Figure 2, conventional metrics like ID accuracy and statistical measures like sparsity showed weak and inconsistent correlations with OOD performance. In contrast, several geometric measures, particularly participation ratio, effective dimension and effective utility, were strong predictors, and effective dimension consistently performs well across all tested architectures.

### 3.2 Findings hold across sizes, optimizers, and datasets

Next, we tested the generality of our findings by varying model size (ResNet18 vs. ResNet34/50), optimizer (SGD vs. AdamW), and the OOD dataset (CIFAR-100 vs. ImageNet). The results, shown in Figure 3, remained consistent. Across all these settings, task-relevant geometric signatures of the ID data were systematically predictive of OOD performance, while other metrics were not. Additional results are provided in Appendix C.

### 3.3 Geometry captures non-monotonic profile of OOD dynamics

Finally, we investigated whether these geometric measures could track OOD performance during the course of training. Our fine-tuning experiments on Stanford Cars revealed a complex, non-monotonic learning dynamic, visually detailed in Figure 3**c** (also see Figure 14). While ID accuracy increases monotonically, OOD accuracy often exhibits a "U-shaped" profile, slightly going down early in training before it improves. This initial dip is qualitatively mirrored by a subtle, inverted "hump" in the geometric measures (e.g., Effective Dimension), which slightly expand before their long-term compression. This demonstrates that geometry not only predicts the final outcome but also provides a more faithful marker of the complex dynamics of OOD generalization during training.

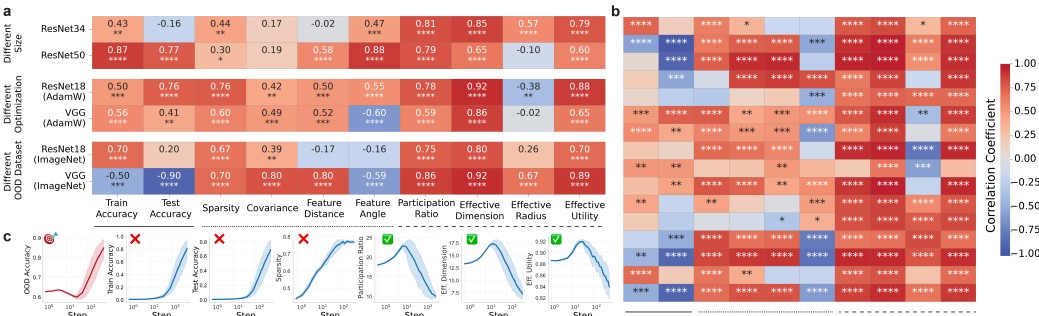

Figure 3: Results across diverse training conditions. **a**: A subset of results compared to those in Figure 2, showing Pearson $r$ values. **b**: A compact summary of all CIFAR hyperparameter sweep results, showing only significance. Numbers indicate Pearson $r$; asterisks denote significance ($* : p \leq 0.05$; $** : p \leq 0.01$; $*** : p \leq 0.001$; $**** : p \leq 0.0001$). x-axis is the same as panel a and y-axis corresponds to different experiment setups. See Figure 4 for the full version. **c**: Optimization trajectories of OOD accuracy and ID measures during fine-tuning experiments on ResNet50 (50 OOD classes) averaged across learning rates. See Figure 14 for individual traces.

## 4 Discussion

In this work, we systematically showed that while conventional metrics fail to robustly predict OOD performance, several geometric measures of ID representations consistently correlate with a model's ability to generalize. These results establish representational geometry as a powerful diagnostic lens for understanding and anticipating OOD failure modes in deep networks.

Our findings open several future directions. First, can these geometric insights provide a deeper mechanistic understanding of generalization? Second, can they inspire new methods for improving

OOD robustness, such as geometry-informed regularization or early-stopping criteria? Third, do these relationships between geometry and generalization extend from artificial models to biological neural systems? Finally, it remains an open question whether the same geometric principles hold across other modalities and task settings, such as language and reinforcement learning.

## Acknowledgments and Disclosure of Funding

We thank Hang Le for the helpful discussion. This work was supported by the Center for Computational Neuroscience at the Flatiron Institute, Simons Foundation. S.C. was partially supported by a Sloan Research Fellowship, a Klingenstein-Simons Award, and the Samsung Advanced Institute of Technology project, "Next Generation Deep Learning: From Pattern Recognition to AI." All experiments were performed using the Flatiron Institute's high-performance computing cluster.

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

# A  Experimental Settings

In this section, we provide a complete description of our experimental setup to facilitate reproducibility.

## A.1  Datasets

We used three primary datasets in our study. For each, we outline its composition, its role in our experiments, and the exact preprocessing and data augmentation pipelines applied.

**CIFAR-10 and CIFAR-100.**  Both are standard image classification benchmarks containing 60,000 color images at a $32 \times 32$ pixel resolution. CIFAR-10 has 10 object categories, while CIFAR-100 has 100 categories that are completely disjoint from those in CIFAR-10. In our primary experiments, CIFAR-10 served as the in-distribution (ID) training dataset, while CIFAR-100 was used as the out-of-distribution (OOD) evaluation set. All images were normalized per-channel using a mean of $(0.4914, 0.4822, 0.4465)$ and a standard deviation of $(0.2023, 0.1994, 0.2010)$.

The preprocessing pipeline differed between training and evaluation. During ID training on CIFAR-10, we applied standard data augmentation, which included padding images with 4 pixels on each side followed by a random $32 \times 32$ crop, as well as random horizontal flipping with a 50% probability. For evaluation on the CIFAR-10 test set and all CIFAR-100 data, these augmentations were omitted. Images were only converted to tensors and normalized, ensuring the evaluation protocol for both ID and OOD data was identical and deterministic.

**ImageNet.**  To test the generality of our findings under a more significant domain shift, we used the ImageNet-1k dataset [24] as an alternative OOD benchmark. This large-scale dataset contains over 1.2 million high-resolution natural images across 1,000 object categories. To ensure compatibility with our CIFAR-trained models, all ImageNet images were first resized to $32 \times 32$ pixels using bicubic interpolation. They were then normalized using the standard ImageNet per-channel mean $(0.485, 0.456, 0.406)$ and standard deviation $(0.229, 0.224, 0.225)$. Consistent with other evaluation protocols, no data augmentation was applied.

**Stanford Cars.**  To study the training dynamics of OOD generalization, we used the Stanford Cars dataset [25], which contains 16,185 images of 196 fine-grained car models. We preserved the dataset's official partition into 8,144 training and 8,041 test images. In addition to this standard example-level split, we introduced a class-level partition by dividing the 196 car models into disjoint in-distribution (ID) and out-of-distribution (OOD) sets. We systematically varied this class partition to create three settings, designating 50, 100, or 150 classes as OOD. The remaining 146, 96, or 46 classes formed the corresponding ID sets. Applying this class partition across the original train/test split yielded four distinct data subsets for each configuration:

1. **ID-train:** Used for fine-tuning the model backbone.
2. **ID-test:** Used for evaluating in-distribution performance of the model.
3. **OOD-train:** Used for training the linear probe.
4. **OOD-test:** Used for evaluating the probe and measuring final OOD accuracy.

To ensure our results were not specific to a particular partition, we generated three unique class splits for each of the three settings using different random seeds, resulting in nine distinct dataset configurations for our fine-tuning experiments. All images were resized to $224 \times 224$ pixels (Lanczos resampling) and normalized with standard ImageNet statistics. During ID fine-tuning, we used random horizontal flips and color jitter for data augmentation; for all evaluations, no augmentations were applied.

## A.2  Model Architectures

We experimented with a diverse set of convolutional neural network (CNN) architectures, covering both classical and modern design principles, to ensure our findings are not tied to a particular

architectural family. For each architecture, we used standard implementations adapted for CIFAR-scale ($32 \times 32$ pixel) inputs. All models were trained from scratch on CIFAR-10 to ensure their representations were learned solely from the ID task, without inherited biases from pre-training.

**ResNet.** We utilized the ResNet family [10], which introduces residual (or "skip") connections to facilitate the training of very deep networks. Our experiments included three variants of increasing capacity: ResNet-18, ResNet-34, and ResNet-50. For CIFAR experiments, we followed convention, using the "basic block" for ResNet-18/34 and the more parameter-efficient "bottleneck block" for ResNet-50. All variants use batch normalization and ReLU activation after each convolution, with downsampling performed via stride-2 convolutions.

**VGG.** We also employed models from the VGG family [11], which represent a classical feedforward design characterized by its simplicity and use of small $3 \times 3$ convolutional filters stacked in sequence. Our implementations of VGG-13, VGG-16, and VGG-19 were augmented with batch normalization after each convolutional layer to improve training stability. The fully connected classifier layers were adapted for CIFAR-scale inputs, with dropout applied for regularization.

**MobileNet.** To include lightweight, efficient architectures, we used MobileNetV1 [13]. Its core innovation is the use of depthwise separable convolutions – a factorization of standard convolutions into a depthwise and a pointwise ($1 \times 1$) step – which dramatically reduces computational cost and parameter count. The architecture begins with a $3 \times 3$ convolution producing 32 channels, followed by a sequence of depthwise–pointwise blocks with progressively increasing channels and optional stride-2 layers for downsampling. A global average pooling layer produces the final feature vector before classification.

**EfficientNet.** We included EfficientNet-B0 [12], a modern architecture that employs a compound scaling method to jointly optimize network depth, width, and resolution. The network is constructed from mobile inverted bottleneck blocks (MBConv) featuring squeeze-and-excitation optimization. Our implementation follows the specific configuration for B0, which is defined by seven stages with the following parameters: block expansions $[1, 6, 6, 6, 6, 6, 6]$, output channels $[16, 24, 40, 80, 112, 192, 320]$, kernel sizes $[3, 3, 5, 3, 5, 5, 3]$, and strides $[1, 2, 2, 2, 1, 2, 1]$. We used a dropout rate of $0.2$ before the final classifier and a stochastic depth (drop connect) schedule that linearly increased to a rate of $0.2$.

**DenseNet.** Finally, we used DenseNet [14], an architecture designed to maximize feature reuse and improve gradient flow by connecting each layer to every other layer within a dense block. Our model follows the CIFAR-specific DenseNet-BC structure, which utilizes 4 dense blocks separated by transition layers that perform $1 \times 1$ convolution and $2 \times 2$ average pooling. Each dense block is composed of bottleneck layers with a $1 \times 1$ convolution followed by a $3 \times 3$ convolution. We used a growth rate of 12 and block configuration $[6, 12, 24, 16]$, resulting in a compact yet expressive network suitable for CIFAR-scale inputs.

This architectural diversity—from deep residual networks to lightweight mobile models and densely connected designs—ensures that our analysis covers a broad range of representational geometries and computational regimes.

## A.3 Optimization and Training Protocols

All models were trained using the cross-entropy loss in PyTorch. Our study employed two distinct training protocols to investigate OOD generalization from different perspectives: a broad hyperparameter sweep on models trained from scratch, and an analysis of the training dynamics during fine-tuning.

**Regime 1: Hyperparameter Sweep (Training from Scratch).** To assess how OOD performance varies across a wide range of final model states, we trained all architectures from scratch on CIFAR-10. We used two optimizers: SGD with a momentum of 0.9, and AdamW [15]. We ran training for 200 epochs with a cosine annealing learning rate schedule, which smoothly decays the learning rate to zero, stabilizing late-stage representation geometry.

For each architecture and optimizer pair, we performed a systematic $4 \times 4$ grid search over the initial learning rate ($\eta_0$) and weight decay ($\lambda$). The specific values for each grid, which were tailored to each architecture family based on empirical best practices, are detailed in Table 1 and Table 2. This diverse grid was designed to produce models in various training regimes, from under- to over-regularized, allowing us to find cases where ID performance is stable while OOD performance varies—a key aspect of our analysis.

| Architecture | Initial learning rate list | Weight decay list |
|---|---|---|
| VGG (13/16/19) | [0.01000, 0.00333, 0.00111, 0.00037] | [0.0010000, 0.0003333, 0.0001111, 0.0000370] |
| ResNet (18/34/50) | [1.00000, 0.50000, 0.25000, 0.12500] | [0.0002000, 0.0001000, 0.0000500, 0.0000250] |
| DenseNet121 | [0.05000, 0.01667, 0.00556, 0.00185] | [0.0005000, 0.0001667, 0.0000556, 0.0000185] |
| MobileNet | [0.20000, 0.06667, 0.02222, 0.00741] | [0.0001000, 0.0000333, 0.0000111, 0.0000037] |
| EfficientNetB0 | [0.20000, 0.06667, 0.02222, 0.00741] | [0.0001000, 0.0000333, 0.0000111, 0.0000037] |

Table 1: Hyperparameter grid for SGD optimizer.

| Architecture | Initial learning rate list | Weight decay list |
|---|---|---|
| VGG (13/16/19) | [0.02000, 0.00500, 0.00125, 0.00031] | [0.0100000, 0.0033333, 0.0011111, 0] |
| ResNet (18/34/50) | [0.10000, 0.02500, 0.00625, 0.00156] | [0.0100000, 0.0050000, 0.0025000, 0] |
| DenseNet121 | [0.05000, 0.02500, 0.01250, 0.00625] | [0.0100000, 0.0033333, 0.0011111, 0] |
| MobileNet | [0.02000, 0.00500, 0.00125, 0.04000] | [0.0100000, 0.0033333, 0.0011111, 0] |
| EfficientNetB0 | [0.10000, 0.05000, 0.01000, 0.00100] | [0.0010000, 0.0003333, 0.0001111, 0] |

Table 2: Hyperparameter grid for AdamW optimizer.

**Regime 2: Fine-tuning Dynamics.**  To investigate how ID representations and OOD performance co-evolve during training, we conducted a separate set of experiments. For these, we fine-tuned an ImageNet pre-trained ResNet50 on our custom ID/OOD splits of the Stanford Cars dataset.

Fine-tuning was performed on the **ID-train** split for 6 epochs using the AdamW optimizer. Instead of a grid search, we tested 10 different logarithmically-spaced learning rates (from $10^{-5}$ to $3.16 \cdot 10^{-4}$) and used a constant learning rate schedule to isolate the effect of training time. During each run, we saved 25 model checkpoints at power-law-spaced steps to capture the early training phase in high resolution. The primary goal of this protocol was to analyze the trajectory of representations over time, not necessarily to achieve maximum final performance.

### A.4   Computing resources

All experiments were conducted on NVIDIA H100 (80GB) or A100 (80GB) GPUs, paired with a 128-core Rome CPU and 1 TB of RAM. Training each model for 200 epochs required approximately 1–3 hours, depending on the architecture and optimizer. Unless otherwise specified, all experiments were run on a single GPU worker. These specifications, together with the full training configurations described in earlier subsections, are provided to facilitate reproducibility.

## B   Details on ID Measures

In this section, we define the performance, statistical, and geometric measures used in our analysis. These are computed on the feature representations extracted from models using the in-distribution (ID) training dataset, unless stated otherwise. Our goal is to identify which properties of a model's ID representations can serve as reliable indicators of its out-of-distribution (OOD) generalization capability.

The measures are grouped into three categories: **performance** measures that quantify classification accuracy, **statistical** measures that summarize low-order distributional properties of features, and **geometric** measures that characterize the structure of class-specific feature manifolds. A key distinction is that while statistical metrics typically operate on pooled features, our primary geometric measures are computed on object manifolds – the per-class point clouds in representation space. This

allows them to directly capture properties relevant to classification, such as manifold size, shape, and correlation structure in the representational space.

We first describe how feature representations are extracted and then define each measure in detail.

## B.1 Representation Extraction

All representational measures are computed on feature vectors extracted from the penultimate layer of each network – the final layer before the classification head. This layer captures high-level, task-specialized features that are not yet collapsed into class logits. For convolutional networks, the feature vector is obtained via global average pooling. The exact layers used for each architecture are listed in Table 3.

| Architecture | Layer name in PyTorch module |
|---|---|
| VGG13 | `features.34` |
| VGG16 | `features.43` |
| VGG19 | `features.52` |
| ResNet18 | `avgpool` |
| ResNet34 | `avgpool` |
| ResNet50 | `avgpool` |
| DenseNet121 | `avg_pool2d` |
| MobileNet | `avg_pool2d` |
| EfficientNetB0 | `adaptive_avg_pool2d` |

Table 3: Exact layer names used for extracting feature representations.

Given an ID dataset $\mathcal{D}_{\mathrm{ID}}$ and a trained network $f_\theta$, let $\mathbf{z}_i \in \mathbb{R}^N$ denote the $N$-dimensional feature vector for the $i$-th input sample $\mathbf{x}_i$ in $\mathcal{D}_{\mathrm{ID}}$, extracted from the layer listed in Table 3. All statistical and geometric measures described in the following subsections are computed from the collection $\{\mathbf{z}_i\}_{i=1}^M$ of such feature vectors, where $M$ is the total number of samples in $\mathcal{D}_{\mathrm{ID}}$.

For measures that require class-specific statistics (e.g., within-class covariance, manifold radius), we further partition $\{\mathbf{z}_i\}$ by ground-truth label into $\{\mathbf{z}_i^\mu\}_{i=1}^{M^\mu}$ for each class $\mu \in \{1, \ldots, P\}$, where $M^\mu$ is the number of samples in class $\mu$.

## B.2 Performance measures

**ID Train and Test Accuracy.** These are the standard top-1 classification accuracies on the ID training and test sets, respectively. They are computed with the full network in evaluation mode and quantify the model's ability to fit and generalize within the training distribution.

**OOD Linear Probe Accuracy.** To assess the transferability of the learned representations to novel classes, we employ a linear probing protocol [21, 20]. In this procedure, the trained network backbone is frozen, and a new linear classifier is trained on top of its features using data from the OOD dataset. This evaluates the intrinsic quality of the representations for OOD tasks without altering the backbone itself. Because our study involves two distinct experimental regimes, we used two corresponding probing protocols.

For the **hyperparameter sweep experiments (CIFAR/ImageNet)**, where the goal was to obtain a stable performance estimate for a final, trained model, we used a sub-sampling approach. We constructed OOD "mini-tasks" by randomly sampling 10 classes from the full OOD dataset. The final reported OOD accuracy is the average performance across 30 such trials, each with an independently drawn set of 10 classes. The linear classifier for each trial was trained using the Adam optimizer with a decaying learning rate and early stopping.

For the **fine-tuning dynamics experiments (Stanford Cars)**, where the goal was to track performance over time for a fixed OOD task, we used a direct, full-dataset approach. For each saved checkpoint, we trained a single linear probe on the entire "OOD-train" split (containing 50, 100, or 150 classes, depending on the configuration). This probe was trained for 100 epochs with a constant learning rate and evaluated on the corresponding full "OOD-test" set.

## B.3 Statistical metrics

We compute a set of statistical descriptors from the ID feature representations to quantify basic structural properties of the learned embedding space. All metrics are computed from the collection of penultimate-layer feature vectors $\{\mathbf{z}_i\}_{i=1}^{M}$ extracted from the ID dataset (see Table 3).

**Activation sparsity.** The activation sparsity measures the proportion of non-zero entries across all feature vectors,

$$\text{sparsity} = \frac{1}{MN} \sum_{i=1}^{M} \sum_{j=1}^{N} \mathbf{1}(|z_{ij}| > \varepsilon),$$

where $N$ is the feature dimension and $\varepsilon = 10^{-6}$ is a small threshold to account for numerical noise. Higher sparsity indicates more silent units on average across the dataset.

**Covariance magnitude.** We compute the empirical covariance matrix $\boldsymbol{\Sigma} \in \mathbb{R}^{N \times N}$ over features and take the mean absolute value of its off-diagonal entries,

$$\text{mean\_covariance} = \frac{2}{N(N-1)} \sum_{j<k} |\Sigma_{jk}|,$$

which reflects the average degree of linear correlation between distinct feature dimensions.

**Pairwise distance.** We compute the mean Euclidean distance between all pairs of feature vectors,

$$\text{mean\_distance} = \frac{2}{M(M-1)} \sum_{i<j} \|\mathbf{z}_i - \mathbf{z}_j\|_2,$$

providing a coarse measure of spread in the representation space.

**Pairwise angle.** After $\ell_2$-normalizing each feature vector, we compute cosine similarities and convert them to angles in radians via $\theta_{ij} = \arccos(\cos\_sim_{ij})$. The mean pairwise angle reflects the typical directional separation between features.

All statistical metrics are computed on the raw feature vectors without centering unless required by the measure (e.g., covariance).

## B.4 Geometric measures: participation ratio and GLUE-based task-relevant metrics

Unlike the statistical measures described above, our geometric analysis operates on *object manifolds*—point clouds in feature space containing activations from the same class. This distinction is important: geometric metrics explicitly quantify per-class representational structure, whereas most statistical metrics aggregate across the entire dataset without regard to class boundaries.

**Participation ratio (PR).** As a conventional baseline for manifold dimensionality, we compute the *participation ratio* (PR) of the penultimate-layer features for each class. Let $\{\mathbf{z}_i^{\mu}\}_{i=1}^{M^{\mu}}$ denote the $M^{\mu}$ feature vectors for the $\mu$-th class, and $\lambda_i^{\mu}$ be the eigenvalues of their covariance matrix. The PR of this class is defined as

$$D_{\mathsf{PR}}^{\mu} = \frac{(\sum_i \lambda_i^{\mu})^2}{\sum_i (\lambda_i^{\mu})^2}, \tag{1}$$

which measures the effective number of principal components with substantial variance. In all figures we present the average of PR over all classes, i.e., $\frac{1}{P} \sum_{\mu} D_{\mathsf{PR}}^{\mu}$. While PR is widely used, it is *task-agnostic* and does not incorporate information about class separability.

### B.4.1 Task-relevant geometric measures from GLUE

To capture the aspects of representational geometry most relevant for classification, we employ the effective geometric measures introduced in the *Geometry Linked to Untangling Efficiency* (GLUE) framework [22], grounded in manifold capacity theory [22, 23].

Analogous to support vector machine (SVM) theory—where an analytical connection between the max-margin linear classifier and its support vectors is used to assess separability in the *best-case sense*—GLUE establishes a similar analytical connection in an *average-case sense*, as follows. Rather than analyzing the max-margin classifier directly in the original $N$-dimensional feature space $\mathbb{R}^N$, GLUE considers random projections to an $N'$-dimensional subspace and evaluates whether the representations remain linearly separable. Intuitively, if the data are highly separable in $\mathbb{R}^N$, they will, with high probability, remain separable even after projection to a much lower $N'$. Conversely, if the data are barely separable in $\mathbb{R}^N$, the probability of maintaining separability will rapidly drop to zero as $N'$ decreases.

Formally, following the modeling and notation in GLUE, each object manifold is modeled as the convex hull of all representations corresponding to the $\mu$-th class:

$$\mathcal{M}^\mu = \text{conv}\left(\{\mathbf{z}_i^\mu\}_{i=1}^M\right),$$

where $\{\mathbf{z}_i^\mu\}$ is the collection of $M$ feature vectors of the $\mu$-th class. A dichotomy vector $\mathbf{y} \in \{-1, 1\}^P$ and a collection $\mathcal{Y} \subset \{-1, 1\}^P$ are chosen by the analyst. Common choices are $\mathcal{Y}$ being the set of all 1-vs-rest dichotomies (e.g., $(1, -1, -1, \ldots, -1), (-1, 1, -1, \ldots, -1), \ldots, (-1, -1, -1, \ldots, 1)$) or $\mathcal{Y} = \{-1, 1\}^P$.

The key quantity in GLUE for measuring the degree of (linear) separability of manifolds is the *critical dimension*, defined as the smallest $N'$ such that the probability of (linear) separability after projection to a random $N'$-dimensional subspace is at least $0.5$:

$$N_{\text{crit}} := \min_{p(N') \geq 0.5} N',$$

where

$$p(N') := \Pr_{\Pi:\mathbb{R}^N \to \mathbb{R}^{N'}}\left[\exists\, \mathbf{w} \in \mathbb{R}^{N'} \text{ s.t. } y^\mu \langle \mathbf{w}, \mathbf{x}^\mu \rangle \geq 0,\ \forall \mu,\ \mathbf{x}^\mu \in \mathcal{M}^\mu\right].$$

By scaling $N_{\text{crit}}$ with the number of manifolds, we define the *classification capacity* $\alpha := P/N_{\text{crit}}$, which intuitively captures the maximal load a network can handle. Larger $\alpha$ corresponds to more separable manifolds in the average-case sense.

GLUE theory relates $\alpha$ to manifold structure through:

$$\alpha = P \cdot \left( \mathop{\mathbb{E}}_{\substack{\mathbf{y}\sim\mathcal{Y} \\ \mathbf{t}\sim\mathcal{N}(0,I_N)}} \left[ \max_{\lambda_i^\mu \geq 0\ \forall \mu, i} \left( \frac{\langle \mathbf{t}, y^\mu \lambda_i^\mu \mathbf{z}_i^\mu \rangle}{\left\| \sum_{\mu,i} y^\mu \lambda_i^\mu \mathbf{z}_i^\mu \right\|_2} \right)^2 \right] \right)^{-1}. \tag{2}$$

Equation 2 can be numerically estimated using a quadratic programming solver (see Algorithm 1 in [22]).

Observe that one can view the optimal solution $\lambda^\mu(\mathbf{y}, \mathbf{t})$ for the inner maximization problem as a function of $\mathbf{y}, \mathbf{t}$. This naturally leads to the following definition of *anchor point* for class $\mu$ as:

$$\mathbf{s}^\mu(\mathbf{y}, \mathbf{t}) := \frac{\sum_i \lambda_i^\mu(\mathbf{y}, \mathbf{t}) \mathbf{z}_i^\mu}{\sum_i \lambda_i^\mu(\mathbf{y}, \mathbf{t})},$$

and stacking them into a matrix $\mathbf{S} \in \mathbb{R}^{P \times N}$, GLUE yields an equivalent form:

$$\alpha = P \cdot \left( \mathop{\mathbb{E}}_{\substack{\mathbf{y}\sim\mathcal{Y} \\ \mathbf{t}\sim\mathcal{N}(0,I_N)}} \left[ (\mathbf{St})^\top (\mathbf{SS}^\top)^\dagger (\mathbf{St}) \right] \right)^{-1}, \tag{3}$$

where $\dagger$ denotes the pseudoinverse. This parallels SVM theory, where the margin is linked to a simple function on the support vectors.

**Center–axis decomposition of anchor points.** For each $\mu \in [P]$, define the anchor center of the $\mu$-th manifold as:

$$\mathbf{s}_0^\mu := \mathbb{E}_{\mathbf{y},\mathbf{t}}\left[\mathbf{s}^\mu(\mathbf{y}, \mathbf{t})\right],$$

and for each $(\mathbf{y}, \mathbf{t})$, define the axis component of the $\mu$-th anchor point as:

$$\mathbf{s}_1^\mu(\mathbf{y}, \mathbf{t}) := \mathbf{s}^\mu(\mathbf{y}, \mathbf{t}) - \mathbf{s}_0^\mu.$$

Similar to $\mathbf{S}$, we denote $\mathbf{S}_0, \mathbf{S}_1(\mathbf{y}, \mathbf{t}) \in \mathbb{R}^{P \times N}$ as the matrices containing $\mathbf{s}_0^\mu$ and $\mathbf{s}_1^\mu(\mathbf{y}, \mathbf{t})$ on their rows, respectively.

With these, define:

$$a(\mathbf{y}, \mathbf{t}) = (\mathbf{St})^\top (\mathbf{SS}^\top)^\dagger (\mathbf{St}),$$

$$b(\mathbf{y}, \mathbf{t}) = (\mathbf{S}_1(\mathbf{y}, \mathbf{t})\mathbf{t})^\top \left(\mathbf{S}_1(\mathbf{y}, \mathbf{t})\mathbf{S}_1(\mathbf{y}, \mathbf{t})^\top\right)^\dagger (\mathbf{S}_1(\mathbf{y}, \mathbf{t})\mathbf{t}),$$

$$c(\mathbf{y}, \mathbf{t}) = (\mathbf{S}_1(\mathbf{y}, \mathbf{t})\mathbf{t})^\top \left(\mathbf{S}_0\mathbf{S}_0^\top + \mathbf{S}_1(\mathbf{y}, \mathbf{t})\mathbf{S}_1(\mathbf{y}, \mathbf{t})^\top\right)^\dagger (\mathbf{S}_1(\mathbf{y}, \mathbf{t})\mathbf{t}).$$

Note that $\alpha = P / \mathbb{E}_{\mathbf{y}, \mathbf{t}}[a(\mathbf{y}, \mathbf{t})]$.

**Effective geometric measures.** GLUE further decomposes $\alpha$ into three measures:

$$\alpha = \Psi_{\text{eff}} \cdot \frac{1 + R_{\text{eff}}^{-2}}{D_{\text{eff}}},$$

where:

- **Effective dimension:**

$$D_{\text{eff}} := \frac{1}{P} \mathbb{E}_{\mathbf{y}, \mathbf{t}}[b(\mathbf{y}, \mathbf{t})]$$

Intuitively, $D_{\text{eff}}$ measures the intrinsic dimensionality of the manifolds while incorporating *axis alignment* between them. Lower $D_{\text{eff}}$ corresponds to more compact, better-aligned manifolds, improving linear separability.

- **Effective radius:**

$$R_{\text{eff}} := \sqrt{\frac{\mathbb{E}_{\mathbf{y}, \mathbf{t}}[c(\mathbf{y}, \mathbf{t})]}{\mathbb{E}_{\mathbf{y}, \mathbf{t}}[b(\mathbf{y}, \mathbf{t}) - c(\mathbf{y}, \mathbf{t})]}}$$

Intuitively, $R_{\text{eff}}$ quantifies the scale of manifold variation relative to its center, incorporating *center alignment* between classes. Smaller $R_{\text{eff}}$ reflects tighter clustering of features around class centers, reducing manifold overlap.

- **Effective utility:**

$$\Psi_{\text{eff}} := \frac{\mathbb{E}_{\mathbf{y}, \mathbf{t}}[c(\mathbf{y}, \mathbf{t})]}{\mathbb{E}_{\mathbf{y}, \mathbf{t}}[a(\mathbf{y}, \mathbf{t})]}$$

Intuitively, $\Psi_{\text{eff}}$ measures the combined effect of *signal-to-noise ratio* (SNR) on separability. Higher $\Psi_{\text{eff}}$ corresponds to manifolds that are both low-dimensional and compact relative to inter-class distances.

For further derivations, illustrations, and examples, see the supplementary materials of [22]. In all our experiments, for each manifold we subsample to 50 points, conduct GLUE analysis on each manifold pair, and apply Gaussianization preprocessing [26] to ensure initial linear separability.

$$\rho_{\mu, \nu}^c := |\langle \mathbf{s}_0^\mu, \mathbf{s}_0^\nu \rangle|$$

$$\rho_{\mu, \nu}^a := \mathbb{E}_{\mathbf{y}, \mathbf{t}}\left[|\langle \mathbf{s}_1^\mu(\mathbf{y}, \mathbf{t}), \mathbf{s}_1^\nu(\mathbf{y}, \mathbf{t}) \rangle|\right]$$

$$\psi_{\mu, \nu} := \mathbb{E}_{\mathbf{y}, \mathbf{t}}\left[|\langle \mathbf{s}_0^\mu, \mathbf{s}_1^\nu(\mathbf{y}, \mathbf{t}) \rangle|\right]$$

# C   Additional Results

In this section, we present supplementary figures that provide a more detailed view of the main findings. A guide to the content of each figure is provided in Table 4.

**Quantification of Relationships.**   We quantify the relationship between ID measures and OOD performance by computing the Pearson correlation coefficient ($r$) and its associated $p$-value via ordinary least-squares linear regression between the measure values and OOD accuracies. For all figures with heatmaps, we annotate each $r$-value with significance asterisks based on its $p$-value: $p \le 0.0001$ (***), $p \le 0.001$ (****), $p \le 0.01$ (**), and $p \le 0.05$ (*).

| Figure label | Model set | Optimizer | ID split | OOD dataset |
| --- | --- | --- | --- | --- |
| Figure 6 | Five DNNs | SGD | Train | CIFAR-100 |
| Figure 7 | Five DNNs | SGD | Test | CIFAR-100 |
| Figure 8 | Five DNNs | AdamW | Train | CIFAR-100 |
| Figure 9 | Five DNNs | AdamW | Test | CIFAR-100 |
| Figure 10 | Three ResNets + Three VGGs | SGD | Train | CIFAR-100 |
| Figure 11 | Three ResNets + Three VGGs | SGD | Test | CIFAR-100 |
| Figure 12 | ResNet18 + VGG19 | SGD | Train | ImageNet subset |
| Figure 13 | ResNet18 + VGG19 | SGD | Test | ImageNet subset |
| Figure 14 | ResNet50 | AdamW | Train | Stanford Cars holdout |

Table 4: Organization of figures in Appendix C. Each row specifies the model set, optimizer, dataset split, and OOD dataset.

**Data Splits for Measures.** The terms "Test" and "Train" in the figure labels indicate whether the representational measures were computed on the ID test set or the ID training set, respectively. For the fine-tuning dynamics, measures are exclusively computed on the "ID-train" split at each saved checkpoint.

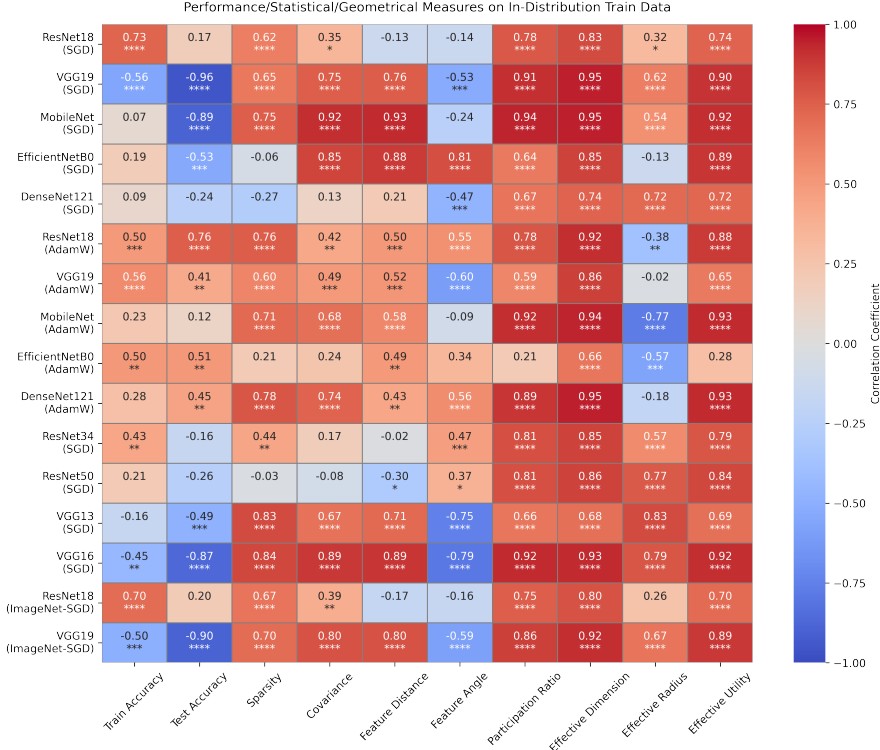

Figure 4: All results, measures computed on the ID *train* set.

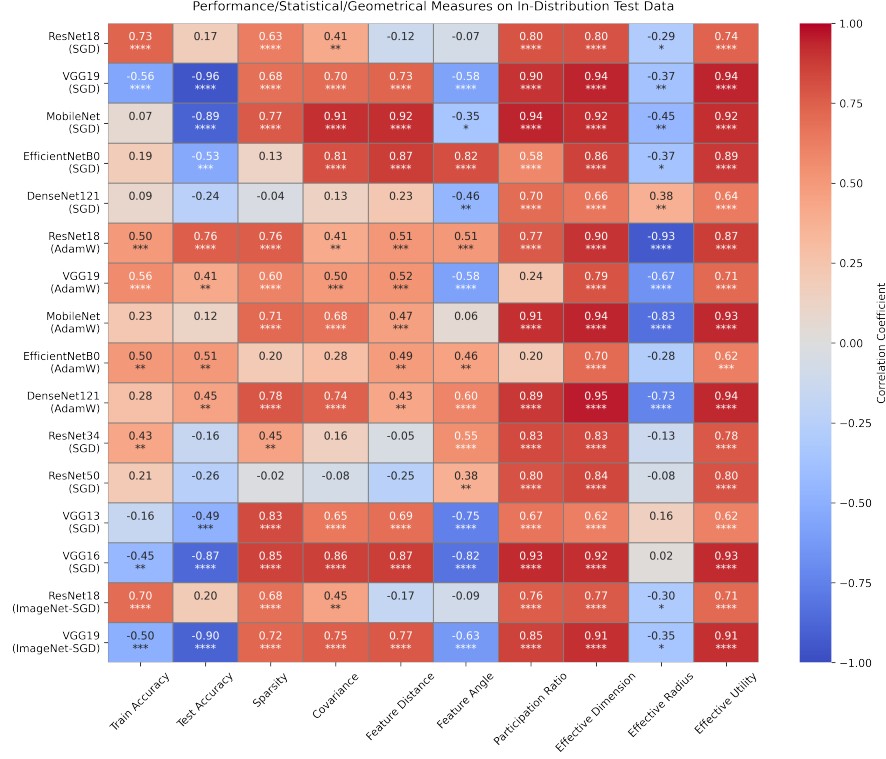

Figure 5: All results, measures computed on the ID *test* set.

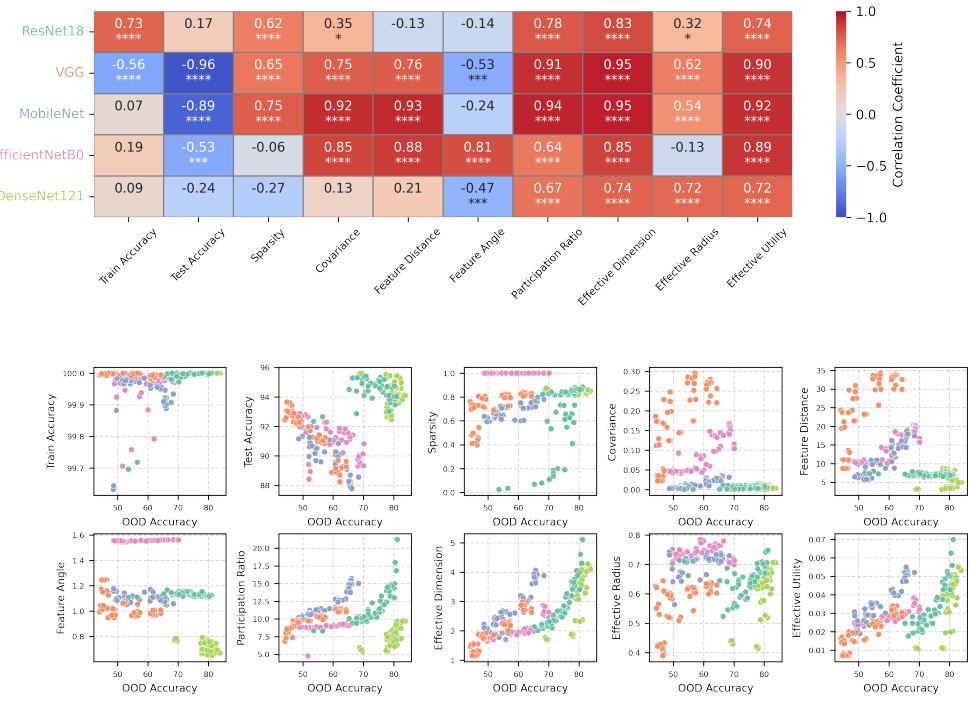

Figure 6: Five DNN architectures, trained with SGD, measures computed on the ID *train* set.

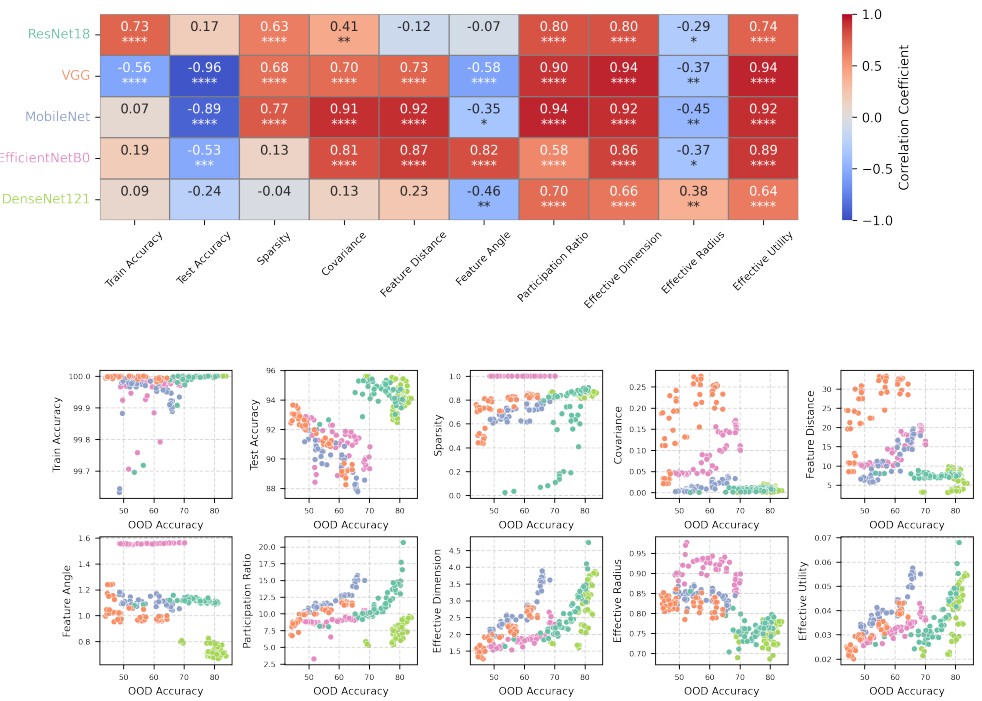

Figure 7: Five DNN architectures, trained with SGD, measures computed on the ID *test* set.

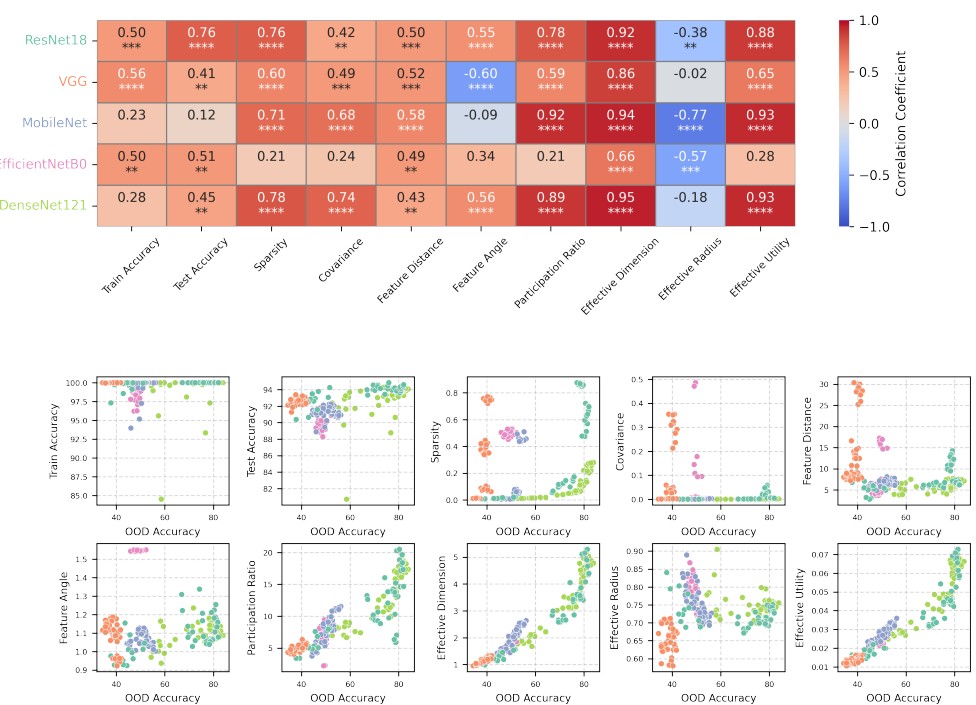

Figure 8: Five DNN architectures, trained with AdamW, measures computed on the ID *train* set.

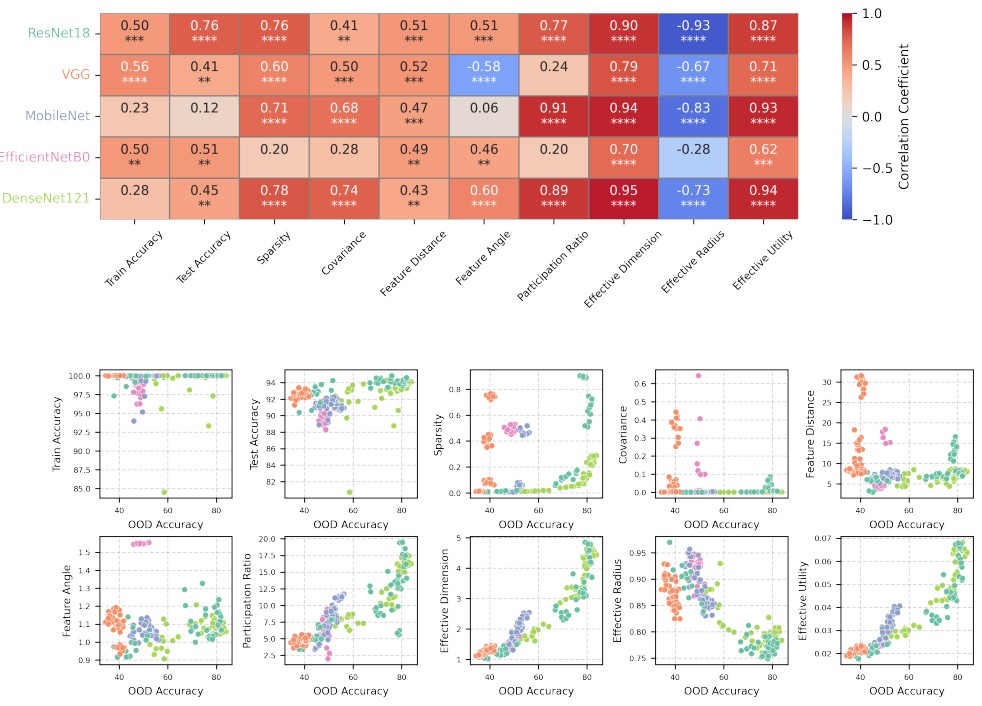

Figure 9: Five DNN architectures, trained with AdamW, measures computed on the ID *test* set.

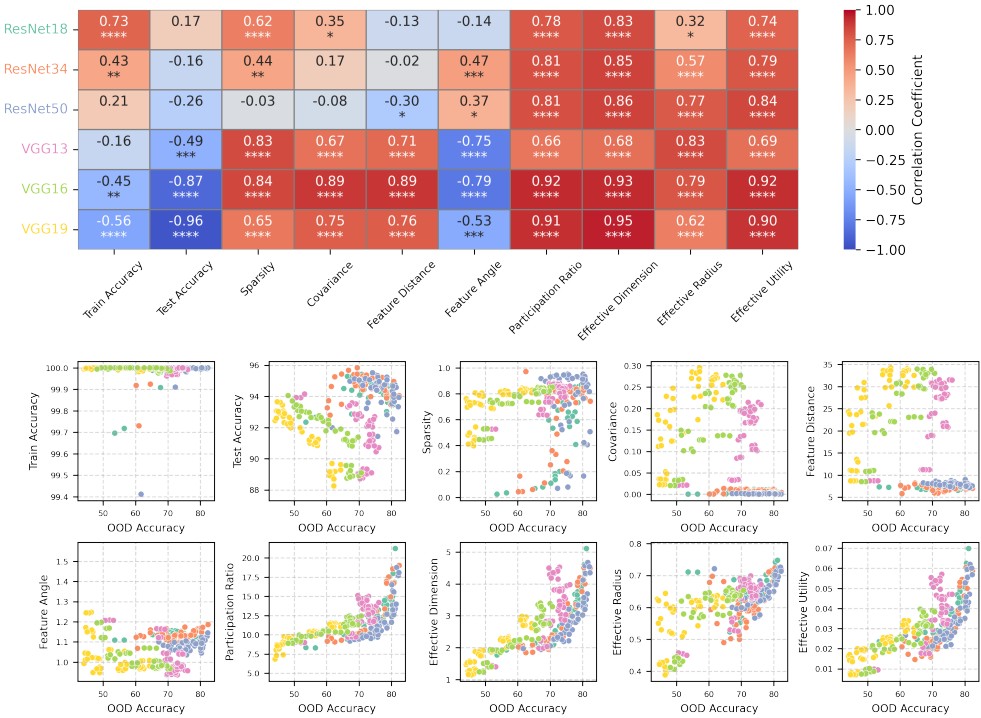

Figure 10: Three ResNet and three VGG architectures, trained with SGD, measures computed on the ID *train* set.

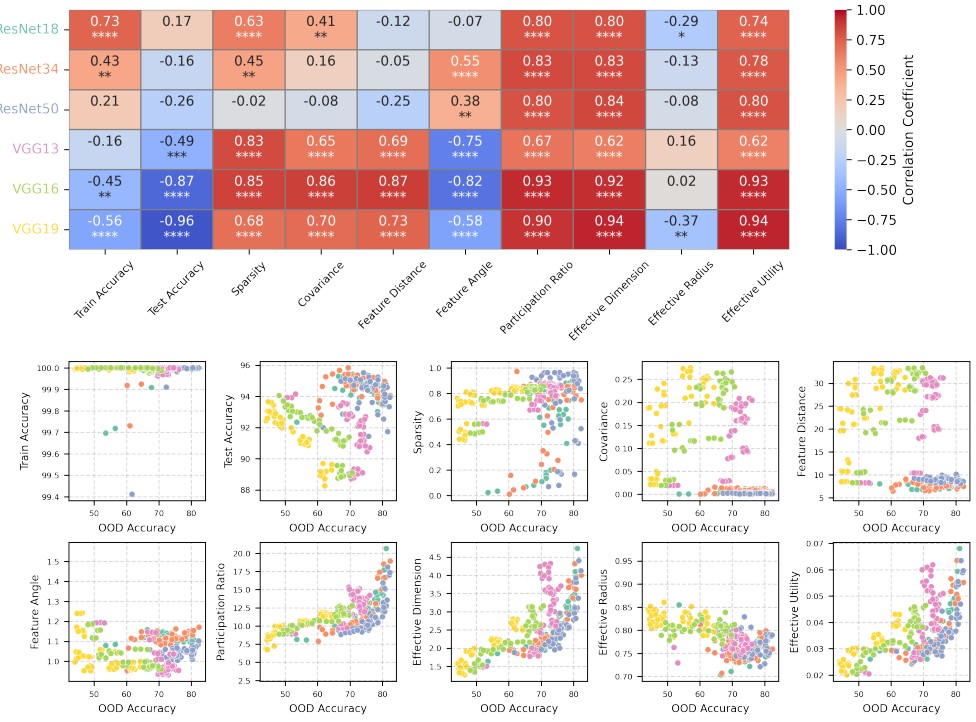

Figure 11: Three ResNet and three VGG architectures, trained with SGD, measures computed on the ID *test* set.

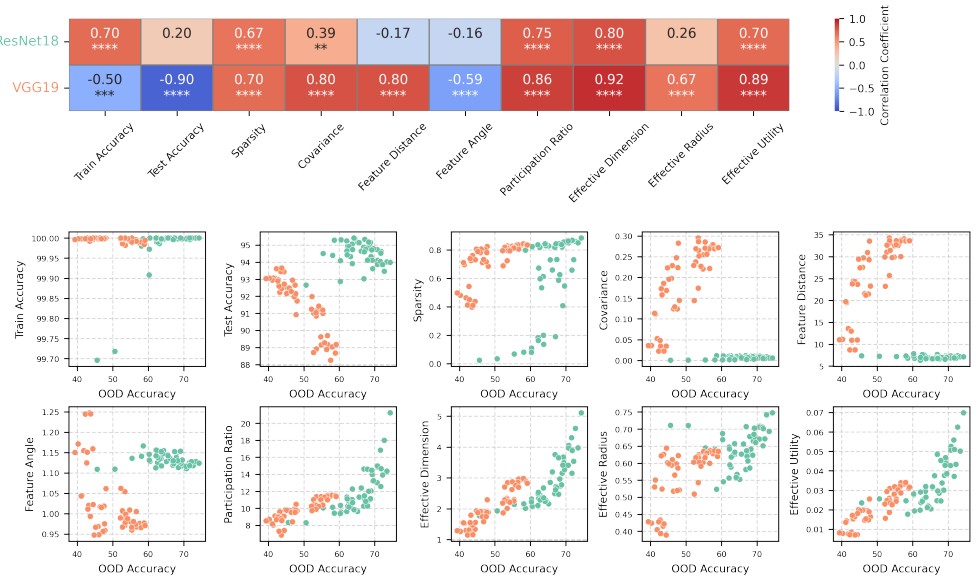

Figure 12: ResNet18 and VGG19, trained with SGD, evaluated on ImageNet subset OOD, measures computed on the ID *train* set.

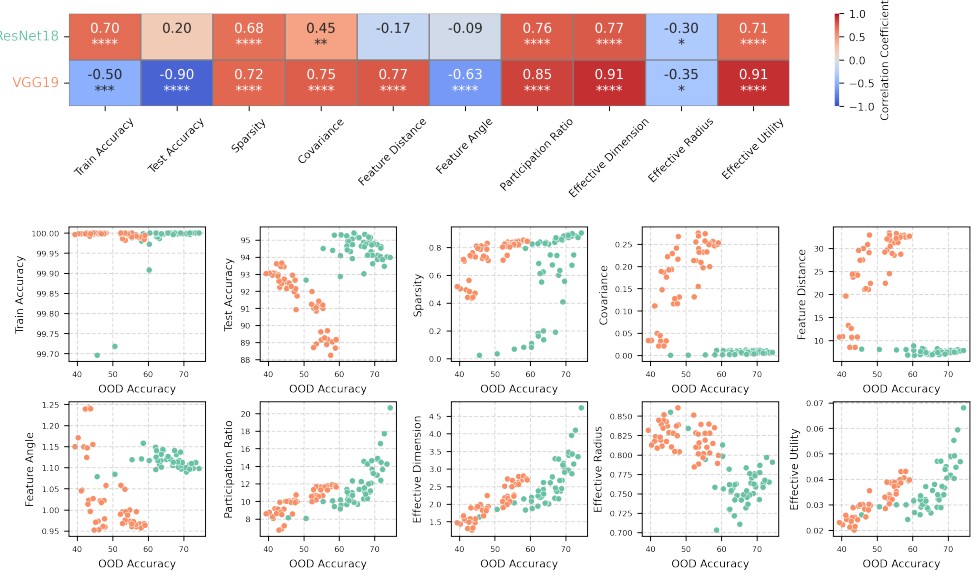

Figure 13: ResNet18 and VGG19, trained with SGD, evaluated on ImageNet subset OOD, measures computed on the ID *test* set.

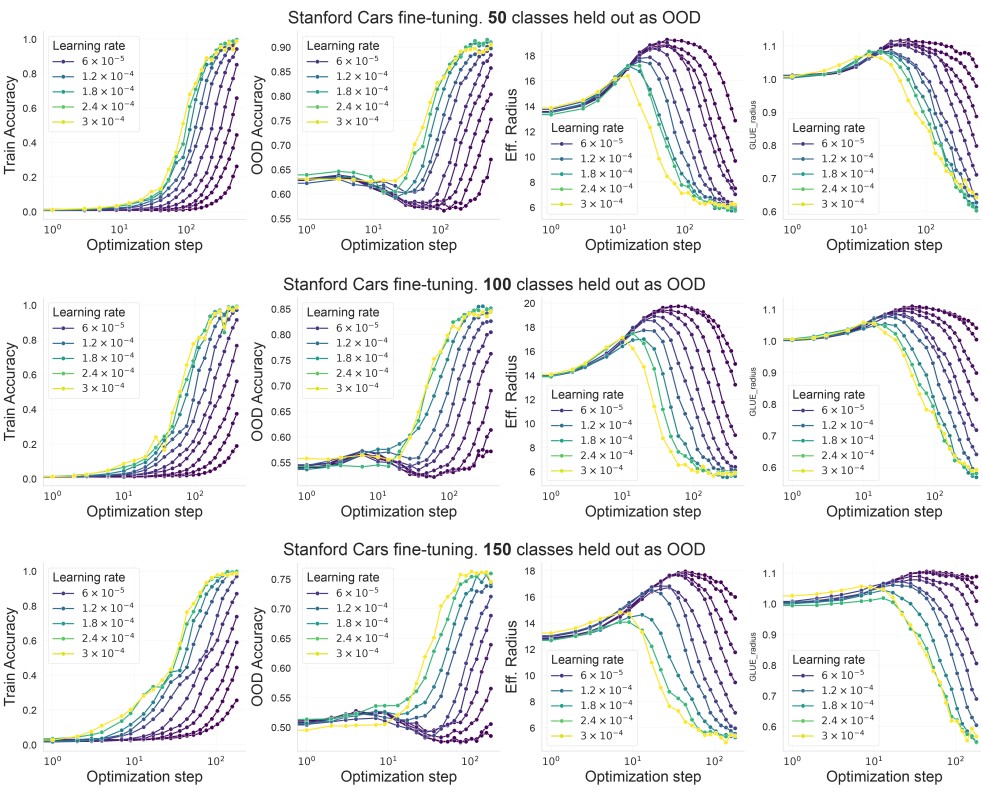

Figure 14: **Training dynamics of ResNet50 fine-tuned on Stanford Cars across different OOD set sizes.** Each row corresponds to a different number of held-out OOD classes. The plots for OOD Accuracy and geometric measures consistently show a non-monotonic profile, where performance initially slightly dips before recovering. This contrasts with the monotonic increase in ID Train Accuracy.

