# OpenReview forum: "Finding Fingerprints of Out-Of-Distribution Failures from In-Distribution Geometry"
_NeurIPS.cc/2025/Workshop/UniReps — UniReps2025_

### Official Review · Reviewer_EEn2 · 2025-09-12
**Very promising results**

**Confidence:** 4

**Review:**

The authors study whether OOD performance can already be predicted by in-distribution (ID) performance. They find that some traditional measures of ID performance have no predictive value for OOD performance, but that several geometric properties of ID performance are highly predictive of OOD performance. This is true across 5 different architectures.

This is a very important line of research as OOD failures are a major issue for modern AI models. As the authors suggest, this work can be used to find "biomarkers" that predict OOD performance before such performance is even tested.

To further test the value of these geometric properties, it would be great if they are computed before OOD generalization is tested and also these should ideally be tested across multiple different OOD datasets. But this work is already very strong for an Extended Abstract submission.

**Score:**

4

**Topic Fit:**

3

---

### Official Review · Reviewer_QXvH · 2025-09-16
**OOD generalization and representational geometry**

**Confidence:** 5

**Review:**

This is an great abstract with a rich set of results on OOD generalization and representational geometry.

The findings resonate with prior work on grokking and geometry (e.g., Zheng et al., UniReps 2024), where linear probing was used to compare grok and standard-trained networks' generalization over training epochs, and a drop in OOD performance in standard-trained network was shown to align with representation compression (see Fig. 4). It would be valuable to cite this result to situate the contribution alongside related findings.

I had two main questions that would love to see followed up:

1. For the OOD evaluation via linear probing, do you expect fine-tuning in this setting to reveal similar trends, or might it behave differently?
2. The effective radius metric seems less correlated with generalization than manifold capacity or effective dimension. Could the authors elaborate on why this might be the case?

Overall, I think this is a strong and promising contribution.

**Score:**

4

**Topic Fit:**

3

---

### Official Review · Reviewer_RwPh · 2025-09-16
**Strong empirical evidence that representational geometry predicts OOD generalization, but limited scope and practical impact.**

**Confidence:** 3

**Review:**

This paper investigates whether geometric properties of in-distribution (ID) feature representations can serve as predictors (“biomarkers”) of out-of-distribution (OOD) generalization performance. The authors systematically evaluate a wide range of CNN architectures, optimizers, and hyperparameters across multiple image datasets. They find that conventional ID performance metrics (train/test accuracy) and simple statistical descriptors (sparsity, covariance) are weak predictors of OOD generalization, while geometric measures of ID manifolds (effective dimension, participation ratio, effective utility from the GLUE framework) strongly correlate with OOD success or failure. Furthermore, the work shows that these geometric measures not only predict final OOD performance but also track its non-monotonic dynamics during training.

Strengths:
1. Clear and systematic experiments across architectures, datasets, and hyperparameters.
2. Empirical evidence that geometry outperforms conventional metrics for OOD prediction.
3. Insightful finding that OOD performance can follow non-monotonic dynamics, captured better by geometry than by accuracy.

Weaknesses:
1. Limited modality and task scope: All experiments are on image classification with CNNs. It is unclear whether the findings extend to other modalities (e.g., NLP, speech, tabular, multimodal) or task settings.
2. Dataset scale and diversity: The primary experiments are on CIFAR-10/100 (32×32 images) with additional results on resized ImageNet and Stanford Cars. While these are standard, they are relatively small-scale compared to real-world distribution shifts (e.g., high-resolution datasets, domain adaptation tasks). Thus, the conclusions may not straightforwardly carry over to large-scale OOD challenges.
3. Correlation, not causation: The paper demonstrates consistent correlations between geometric measures (e.g., effective dimension) and OOD accuracy, but does not fully explain why geometry governs generalization. Without a causal story or ablations showing mechanistic links, the findings risk being treated as descriptive rather than explanatory.
4. Evaluation protocol may be too narrow: OOD generalization is assessed exclusively via linear probing on frozen features. While linear probing is a common diagnostic tool, it does not necessarily reflect performance in real-world deployment where full fine-tuning, domain adaptation, or non-linear heads are used. It remains unclear whether geometry retains the same predictive power under those conditions.

**Score:**

3

**Topic Fit:**

3

---

### Official Review · Reviewer_KDRK · 2025-09-16
**Review of Submission16 by Reviewer KDRK**

**Confidence:** 4

**Review:**

**Summmary**: The authors present experimental results on various image classification tasks, showing that certain geometric measures characterizing the representations obtained on the training data distribution can serve as indicators of degraded performance on out-of-distribution data and non-monotonic learning dynamics during fine-tuning on it.

**Strengths:** Within the limited space of the extended abstract format, the paper provides an excellent summary of the key results and experimental evaluation. The setup and results are presented with clarity. The experimental design and evaluation protocol are appropriate for the stated purpose, and the findings are original, with potential relevance in broader areas, possibly stimulating further discussion. A markedly higher correlation of geometric measures—compared to standard performance and statistical measures—with OOD performance is demonstrated across various benchmark datasets and network architectures. The supplementary material includes detailed descriptions of the employed metrics, experimental settings, hyperparameters, datasets, and architectures, further supporting the reproducibility of the results.

**Weaknesses:** The manuscript may be slightly improved by addressing the following. 1. In Figure 2, the legend from the first subfigure would be better placed underneath the entire subfigure, if space limits permit. In addition, it would be helpful to clearly indicate ID/OOD for train–test accuracy on all figures to avoid misunderstandings.
2. More importantly, the discussion of the results could be expanded by providing potential explanations for the observed patterns—particularly, why the considered geometric measures appear to be stronger predictors of OOD performance, whereas standard statistical metrics (e.g., sparsity) are not.

**Score:**

4

**Topic Fit:**

2